# FEM Based Preliminary Design Optimization in Case of Large Power Transformers

**Tamás Orosz \*** , **David Pánek** and **Pavel Karban**

Department of Theory of Electrical Engineering, University of West Bohemia, Univerzitni 26, 306 14 Pilsen, Czech Republic; panek50@kte.zcu.cz (D.P.); karban@kte.zcu.cz (P.K.)

\* Correspondence: tamas@kte.zcu.cz

**Abstract:** Since large power transformers are custom-made, and their design process is a labor-intensive task, their design process is split into different parts. In tendering, the price calculation is based on the preliminary design of the transformer. Due to the complexity of this task, it belongs to the most general branch of discrete, non-linear mathematical optimization problems. Most of the published algorithms are using a copper filling factor based winding model to calculate the main dimensions of the transformer during this first, preliminary design step. Therefore, these cost optimization methods are not considering the detailed winding layout and the conductor dimensions. However, the knowledge of the exact conductor dimensions is essential to calculate the thermal behaviour of the windings and make a more accurate stray loss calculation. The paper presents a novel, evolutionary algorithm-based transformer optimization method which can determine the optimal conductor shape for the windings during this examined preliminary design stage. The accuracy of the presented FEM method was tested on an existing transformer design. Then the results of the proposed optimization method have been compared with a validated transformer design optimization algorithm.

**Keywords:** design optimization; evolutionary computation; finite element analysis; power transformers

## 1. Introduction

Large power transformers are generally specific, tailored to the unique customer requirements. In case of large machines their design process is a complex, labour intensive task, where many physical fields have to be considered simultaneously [1–3]. During the tendering procedure, a preliminary design is made to determine the final price and the key-design parameters of the cost optimal transformer design (Figure 1). It is important to consider not only the technical feasibility, but the economic aspects, as well. Generally, the total cost of ownership (TOC) is used as a goal function [4] to consider the lifetime costs of the transformer [2,4–8].

The uniqueness is a very important factor during the design and optimization of very large machines. Generally, only one design is built with the given requirements, there is no other possibility to tune or refine the parameters after the measurements. Moreover, the manufacturing cost of these machines are very high, therefore a company can win (or loose) a lot of money if it can won the bidding procedure with a good preliminary design. The mathematical representation of this problem belongs to the most general branch of discrete, non-linear mathematical optimization problems [9]. During the preliminary design process, this good design have to be selected from several thousands of feasible transformer designs, in a very short time (Figure 2). Many different methodologies have been published in the literature, which use a lot of simplifications [10–12]. These can decrease the robustness of the solution, due to the modelling uncertainties [13].

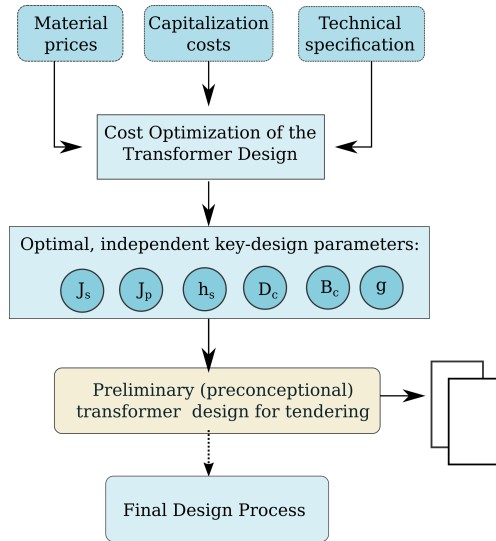

**Figure 1.** Schematic view of a large power transformer design process.

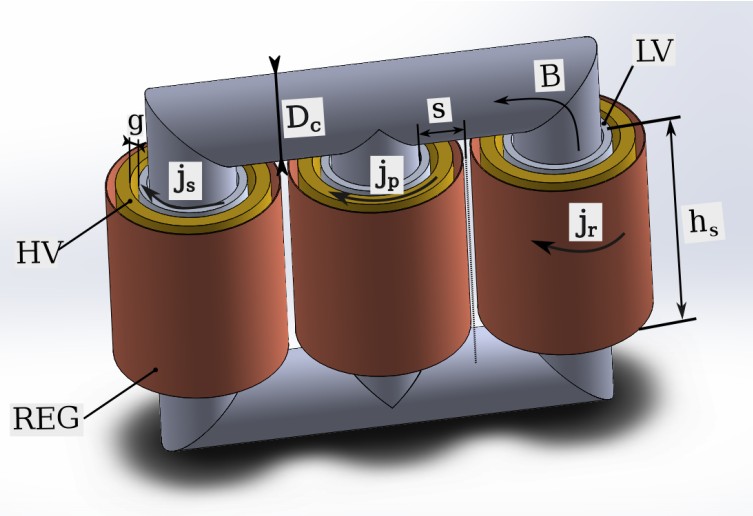

**Figure 2.** Schematic view the optimized key-design parameters of a three phase, core-form large power transformer. All of the optimized, independent key-design parameters have been noted on the picture.

In the case of very large power transformers, several winding layouts are used Figure 3, because of their benefits and drawbacks. However, the nowadays used algorithms replace the detailed winding layouts with copper filling factor based models, or do not consider these differences [10–12,14–19]. The knowledge of the conductor sizes and the winding layout are essential to make an accurate stray loss calculation and create a more robust solution [3,9,20]. Most of the existing algorithms are using copper filling factor based winding models to the optimization [10–12,14,15]. This modelling technique is widely used in the transformer industry, because it estimates well the losses, the outer dimensions of the winding and the related main electrical properties of the transformer [3,9]. Some of them use FEM (Finite Element Method) techniques in their optimization loop to refine the results [11,16,18,19,21]. However, these algorithms uses the FEM method only to refine the best individual from the generation, they are not considers the short-circuit impedance and other important electrical parameters [18] during the calculation.

Since the cost and constraints are generally non-linear functions of the design variables, the mathematical representation of the preliminary transformer design optimization problem is strongly non-linear. There can be several extremums, which has nearly the same TOC and the designer can think that these solutions are very similar. However their key-design parameters can be very different.

Therefore, it is important to check the feasibility of the windings and make precise short-circuit impedance calculation during this very first optimization stage to provide a more robust solution.

This paper proposes an evolutionary algorithm based method, which can calculate the optimal conductor sizes and winding layout in the preliminary optimization stage, to provide more robust key-design parameters for the final design. The analytical part of the transformer model is used to calculate some electrical parameters and the shape of the core and winding system (Figure 3).

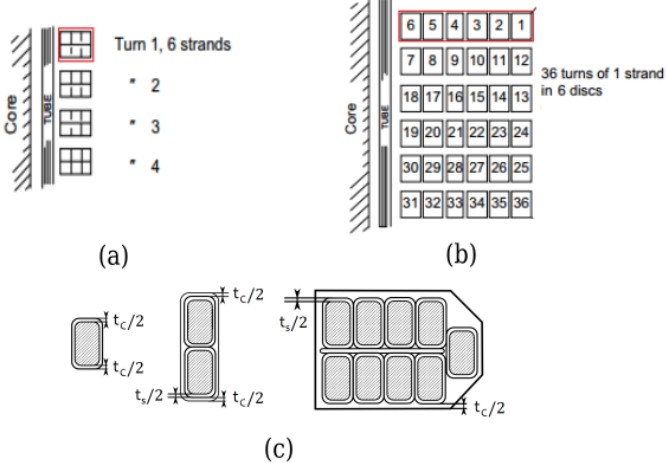

**Figure 3.** Typical winding arrangements: (**a**) a helical winding with 6 parallel strands in one turn (**b**) a disc winding from 1 strand. Typical winding materials (**c**): single conductor, axial twin, continuously transposed cable [1].

Then, the algorithm uses a FEM method directly on every single individual design to calculate the load losses and the short circuit impedance in a sole optimization loop [10–12,14–16]. Finally, an embedded GP model is used to calculate the optimal winding layouts [22,23], which solver checks the proposed layout feasibility and guarantees that the found optimal conductor sizes are the global optimum. The transformer optimization process is realized in the Ārtap framework [24], which tool was developed for robust design analysis and provides the sufficient interfaces, algorithms and FEM solver for the analysis.

## 2. Proposed Methodology

### 2.1. Transformer Model for the Optimization

The transformer is modelled by its active part (the core and the windings). This approximation is widely used in the industry, because its determines well the final dimensions of the transformer [3,10,16]. However, this approximation omits the mass and the cost of the external cooling system and many assemblies, which are generally modeled and designed only in the final design stage. Many transformer models has been published in the literature for preliminary design optimization of power transformers [10,16]. The proposed methodology is based on a widely used model, which was published in the following papers [25–27] and extends this FEM method based calculation to determine the load losses and the short circuit impedance of the transformer. The FEM methodology is used to calculate the magnetic field distribution in the working window of the transformer, which takes the radial part of the magnetic field into account. Moreover, the final, geometric programming based optimization models can calculate the detailed conductor layout for the windings, not only use a copper filling factor based winding model as the previous methods. The proposed geometric programming based equation system can be applied for disc type windings. The other winding types (helical and other special winding types) can be modeled by this method, but their equation system should be derived similarly.

The proposed algorithm can handle one and three phase transformers. The analytical formulation of the proposed algorithm can handle three and five legged transformer cores, as well. The transformer core is modelled by its diameter $D_c$ and its planned filling factor, which takes into consideration the applied manufacturing technology(lamination, number of cooling ducts in the core and the stacking factor). In the paper, the equation system and every calculation is shown on a three phase, three legged transformer with reversing tap-changing method (Figure 2). The realized winding model contains three windings: low voltage (LV), high voltage (HV) and a regulating winding (Reg) (Figure 2). All of the optimized variables are listed in Table 1.

**Table 1.** The parameters of the optimized active part model.

| Quantity | Dimension | Variable |
|---|---|---|
| Independent variables | | |
| Core diameter | mm | $D_c$ |
| Flux density in the core | T | $B$ |
| Main insulation distance | mm | $g$ |
| Current density in the secondary coil | A/mm$^2$ | $j_s$ |
| Current density in the primary coil | A/mm$^2$ | $j_p$ |
| Current density in the regulating coil | A/mm$^2$ | $j_r$ |
| Height of the secondary winding | mm | $h_s$ |
| Dependent parameters (Analytical) | | |
| Width of the working window | mm | $s$ |
| Core mass | t | $M_c$ |
| Radial thickness of secondary winding | mm | $t_s$ |
| Mean radius of secondary winding | mm | $r_s$ |
| Radial thickness of primary winding | mm | $t_p$ |
| Mean radius of primary winding | mm | $r_p$ |
| Radial thickness of regulating winding | mm | $t_r$ |
| Mean radius of regulating winding | mm | $r_r$ |
| No Load Loss | kW | $P_{nll}$ |
| Dependent parameters (FEM) | | |
| Short circuit impedance | % | $SCI$ |
| Maximum of radial flux density in LV | T | $B_{rs}$ |
| Maximum of radial flux density in HV | T | $B_{rp}$ |
| Maximum of axial flux density in LV | T | $B_{as}$ |
| Maximum of axial flux density in HV | T | $B_{ap}$ |
| Dependent parameters (GP sub-problem) | | |
| Number of turns in a winding | # | $n$ |
| Number of conductors in a turn | # | $n_c$ |
| Number of axial turns | # | $n_{ax}$ |
| Number of radial turns | # | $n_{rad}$ |
| Copper area in one turn | mm$^2$ | $A_{cu}$ |
| Copper volume in the winding | mm$^3$ | $V_{cu}$ |
| Copper mass in the winding | kg | $M_k$ |
| Optimal conductor height | mm | $h^*$ |
| Optimal conductor width | mm | $w^*$ |
| Dependent parameters (Complex) | | |
| Load Loss | kW | $P_{ll}$ |
| Total Cost of Ownership | € | $TOC$ |

In the applied methodology, every possible transformer design is represented as an individual. These individuals contains independent and dependent parameters (Table 1), these parameters represents the genes of the individual. Every dependent parameter can be determined by the knowledge of the independent values and the specification. The independent parameters are generated

and optimized by the applied evolutionary algorithm (NSGA-II). The calculation of the dependent parameters are made in every iteration step, by the redefined evaluator function of the optimization framework (Ārtap [24]). This calculation consist of three main calculation steps: the solution of the analytical model, the FEM calculation and the embedded geometric programming based model. The structure of the implemented evaluator function is shown in Algorithm 1. The following subsections show the applied optimization framework [24] and explain the details of the different calculation steps (Figure 4).

---

**Algorithm 1** Transformer Model Evaluator

---

    **function** EVALUATOR(p)    ▷ p means the independent design parameters, which generated by NSGA-II within the given search space
2:        Evaluates the analytical expressions    ▷ determine the main geometrical design parameters

4:        **if** The analytical solution is not feasible  **then**
            **return** TOC = inf
6:
        **end if**
8:        Runs Agros2D – FEM calculation –
            Determine $SCI$ from the magnetic energy
10:          Determine $B_{axp}$, $B_{radp}$ and $B_{axs}$, $B_{rads}$ values

12:        **if** Check SCI is False **then**
            **return** TOC = inf
14:
        **end if**
16:        Runs the GP based winding model
            calculates $h^*$, $w^*$ for both of the windings
18:        calculate the load losses, TOC
        **return** TOC
20: **end function**

---

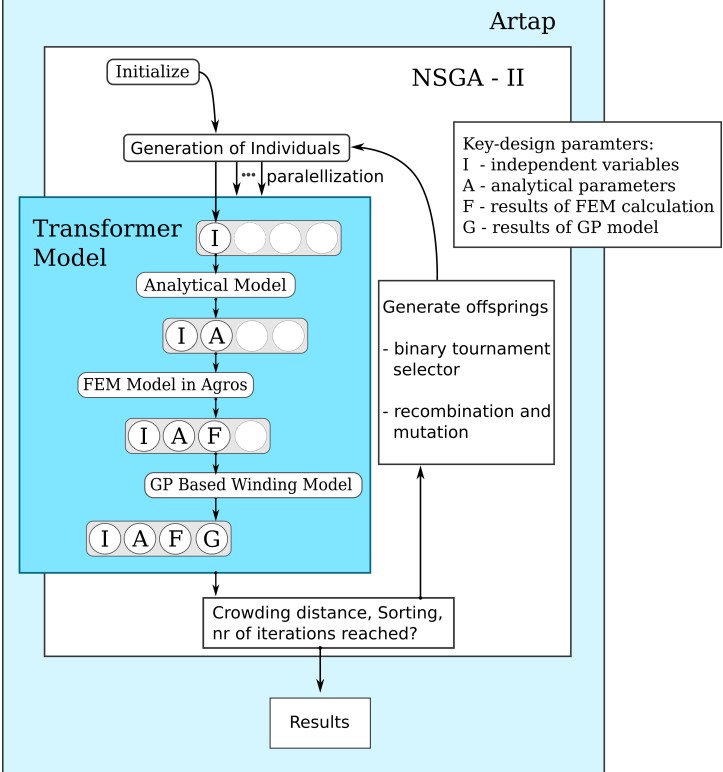

**Figure 4.** Structure of the realized methodology in the framework.

## 2.2. Objective Function—Total Cost of Ownership

The objective function is the total cost of ownership. This function contains the manufacturing cost of the active part and the cost of the calculated losses [2,4,28]:

$$TOC = K_1 \cdot P_{nll} + K_2 \cdot P_{ll} + C_0 \cdot M_C + \sum_{k=1}^{n} C_k \cdot M_k, \tag{1}$$

where TOC is the total cost of ownership of the active part in € and also the objective function of this optimization method. $K_1$ is the capitalized cost of the no-load loss and $K_2$ is the load loss capitalization cost in €/kW. $P_{nll}$ is the no-load loss of the transformer in kW and $P_{ll}$ is the sum of the load losses generated in the active part in kW. $M_k$ is the mass of the $k$th part of the model ($k$ represents the core, LV, HV and Reg windings) in kg and $C_k$ represents the specific cost of the transformer part in €/kg.

## 2.3. FEM Model

The analytical methods generally compute only the axial components of the magnetic field in the working window of a transformer [3,20]. Therefore, those effects, which caused by the radial component of the magnetic field, cannot be considered by the analytical methods. The role of the applied FEM model is to provide a more accurate magnetic field calculation in the working window of the transformer. A 2D, magneto-static FEM method is used for this calculation. This technique originally published and implemented by Andersen [20]. This simple FEM method is widely used in the industry to determine the load losses and the short-circuit forces and impedance of the transformer [3]. Besides its accuracy, the calculation time of one model is within 1 s.

The magnetic core can be defined by its relative permeability ($\mu_r$), it can be some of tens of thousands. During the simulations it was defined as $\mu_r = 10,000$. It can be a number between 10,000 and 50,000 [3]. However it doesn't effect on the solution, because almost all energy is stored in the non-magnetic regions, where $\mu_r = 1$, outside of the core. We can also use the assumption of [20], that the radial component of the magnetic flux density is perpendicular to the core. Other regions, including the windings are defined by $\mu_r = 1$. The magnetic field in the working window of a transformer that is generally nonlinear can be described by the magnetic vector potential $\vec{A}$ in the following form:

$$\Delta \vec{A} = \mu \vec{J}, \tag{2}$$

where $\mu$ denotes the magnetic permeability. Symbol $\vec{J}$ denotes the density of field currents in the windings. The boundary condition along a sufficiently distant boundary is Dirichlet type. The magnetic permeability in every cell of the discretization mesh is assumed constant and corresponds to the corresponding magnetic flux density. By the solution of this problem, the value of the short-circuit reactance can be calculated from the total magnetic energy ($W_m$), evaluated at the peak current ($I_p$) [9,20]:

$$x_L = \frac{4 \cdot f \cdot W_m}{I_p^2} \tag{3}$$

The other result of the calculation is the maximum values of the $B_{ax}$ and $B_z$ values in the windings. These values are used for the load loss calculation in the windings.

## 2.4. Ārtap

Ārtap is an optimization framework developed within University of West Bohemia [24]. Written in Python, it is mainly inspired by projects OpenMDAO [29] and Platypus [30]. Ārtap aims to provide an extensive infrastructure for robust design optimization problems [31–33] in a simple, user friendly way. Moreover, it contains an integrated FEM solver Agros-Suite [34], which is used in this paper for the FEM calculations. These implemented tools offers an easy and straightforward solution for that very frequent engineering problems, where more, different numerical solvers and codes are used to

evaluate one specific solution. Ārtap offers a wide variety of optimization algorithms, some of them coded directly (NSGA-II [35], PSO [36], Eps-Moea [37], etc.) the others can invoked from libraries via wrappers (Bayesopt [38], Nlopt [39] and Scipy [40] libraries). Ārtap offers integrated solutions to directly run FEM solvers from this evaluator function (Agros2D [34], Comsol). The only task of the user is to redefine the evaluator function of Ārtap with the code of the specific calculation. Then Ārtap can solve it automatically with the selected optimization method. Moreover, Ārtap provides automatic parallelisation of the optimization process, like Platypus [30] and PaGMO [41].

*2.5. NSGA-II*

The algorithm NSGA-II (Non-dominated Sorting Genetic Algorithm) was proposed by Deb et al. in 2000 [35], as an improved version of the NSGA algorithm. NSGA-II is one of the most popularly used, genetic algorithm based, multi-objective optimization technique [42]. Due to its following three advantageous characteristics, which were outperformed the existing algorithms when it was published [35]. Firstly, it has a fast, non-dominated sorting approach. The overall computational complexity of this algorithm is almost $O(MN^2)$. Secondly, this algorithm uses elitist strategy, which does not allow to delete some already found Pareto optimal solution. Finally, it has explicit diversity preservation mechanism, which ensures good convergence stability [42]. The pseudo code of the NSGA-II algorithm is shown in Algorithm 2. This is an adopted version of the original pseudo code [35,43], which description already contains the arbitrarily re-definable evaluator function ($f$) of Ārtap.

---

**Algorithm 2** NSGA II

---

1:　**function** NSGAII(n, g, f)　　　　　▷ f means our unique function which calculates TOC and the key design-parameters for an individual
2:　　　initialize parent population ($P$)
3:　　　generate random population ($R$)
4:　　　run f for every individual
5:　　　Sorting, Assign Rank - Pareto dominance -
6:　　　Generate Offsprings ($O$) - next generation
7:　　　　　Binary Tournament Selector
8:　　　　　Recombination and Mutation
9:　　　**for** i := 1 to g **do**　　　　　　　　　　　　　▷ g: max number of generations
10:　　　　　**for** on each $P$ and $O$ in population **do**
11:　　　　　　Sorting, Assign Rank - Pareto dominance -
12:　　　　　　Generate sets of non-dominated vectors
13:　　　　　　Loop – evaluates the user defined $f$ function – and add solutions to next generation starting from the first front until $n$ determine crowding distance between points on each front
14:　　　　　**end for**
15:　　　　　Select individuals (elitist) with lower rank and are outside a crowding distance
16:　　　　　Generate Offsprings ($O$) - next generation
17:　　　　　　Binary Tournament Selector
18:　　　　　　Recombination and Mutation
19:　　　**end for**
20:　**end function**

---

*2.6. Analytical Calculations*

This is the first part of the calculation of the dependent parameters. It uses similar electrical and geometrical formulas, as like the other MDM heuristic based methods to determine the core dimensions, the core losses and the outer dimensions of the windings. This calculation needs a guess for the copper filling factor, which will be replaced with the exact winding layout during the embedded geometric programming part of the algorithm.

## 2.7. Power Criteria in Working Window

$$P_w = 4.44\lambda_c R_c^2 \lambda_w f h_w t_w j_w^2, \tag{4}$$

where $P_w$ means the nominal power of the winding, $\lambda_c$ is the stacking factor of the core, $\lambda_w$ is the copper filling factor of the winding, which used in this first part of the algorithm, to calculate the overall dimensions of the winding. $h_w$ is the height of the winding, $t_w$ is the thickness of the winding, $j_w$ is the current density of the winding.

## 2.8. Regulating Winding Dimensions

The model assumes that the design contains a diverter switch for the regulation. The short circuit impedance is calculated to the nominal state when the regulating winding is de-energized.

$$t_r = \frac{P_{reg}}{j_{reg}^2 \alpha_{reg} h_{in} U_{reg} \lambda_{reg}}, \tag{5}$$

where $t_r$ and $h_{in}$ are the radial thickness and the height of the winding, the $\lambda_{reg}$ means the copper filling factor of the winding.

### Turn Voltage

The turn voltage of the windings is calculated from the given power and the independent variables, the calculation can be formulated in the next form:

$$U_T = 4.44\lambda_c R_c^2 \lambda_{in} f \tag{6}$$

where $U_T$ is the turn voltage in V, $\lambda_c$ is the filling factor of the core.

## 2.9. Core Mass and No-Load Loss Calculation

Similarly to the metaheuristic method in [27], in the case of a three phase three legged core, the core mass can be calculated by the following formulas:

$$M_c = M_{leg} + M_{yoke} + M_{corner}, \tag{7}$$

$$M_{corner} = R_c^3 \cdot \lambda_c \cdot \pi \cdot \rho_{fe} \cdot \zeta, \tag{8}$$

$$M_{column} = R_c^2 \cdot \lambda_c \cdot \pi \cdot \rho_{fe} \cdot (EI_{TOP} + EI_{BOT} + h_{in}), \tag{9}$$

$$M_{yoke} = R_c^2 \cdot \lambda_c \cdot \pi \cdot \rho_{fe} \cdot (4 \cdot s + 2 \cdot p_d + 6 \cdot R_c), \tag{10}$$

where $M_{corner}$ is the mass of the corners of the core, $M_{leg}$ is the mass of the leg, $M_{yoke}$ is the mass of the yoke. $\lambda_c$ is the filling factor of the core, it depends on the quality of the applied electrical steel and the construction technology. $\zeta$ is a technology dependent factor for the core volume calculation, $\rho_{fe}$ is the density of the electrical steel. $EI_{TOP}$ and $EI_{BOT}$ are the end insulation distances, between the bottom and the top of the yoke and the inner winding , $p_d$ is the phase insulation $h_{in}$ represents the height of the inner winding and $s$ represents the width of the working window. The $h_{in}$ winding is used as a reference height in the model as in the metaheuristic method based optimization [27]. The height of the outer and the regulating windings are taken into consideration by a simple multiplication of one factor. Hysteresis ($P_{chyst}$) and eddy current losses ($P_{ceddy}$) cause together the core-losses:

$$P_{nll} = P_{ceddy} + P_{chyst}, \tag{11}$$

In high quality electrical steels, the hysteresis and eddy current losses contribute about equally to the total loss. Eddy current loss, occurring on account of eddy currents produced due to induced voltages in laminations [3,27,44]. Where hysteresis loss is a function of the area of hysteresis loop:

$$P_{ch} = k_1 \cdot f \cdot t^2 \cdot B_p^n \tag{12}$$

where $k_1$ is a material dependent empirical factor, $B_p$ is the peak value of the flux density and $n$ is the Steinmetz constant, which depends on the lamination and the operating flux [3]:

$$P_{ceddy} = k_2 \cdot f^2 \cdot t^2 \cdot B_c^2 \tag{13}$$

where $k_2$ is a material dependent factor, $f$ is the frequency, $t$ is the lamination thickness and $B_c$ is the inductance. These equations describes the theory of the loss generation in the magnetic core. However, this optimization model uses measurement results to determine the core losses. Every manufacturer provides a loss-curve from their steels, where the loss is a function of the induction in W/kg units. These practical formulas are the results of measurements, which are made by an Epstein-apparatus and they are take the hysteresis and eddy losses into account. The applied loss function is fitted to the applied electrical steel data (M1H [45]) and approximated by a polynomial expression [9,14,27,46]:

$$p_{nll} = a_0 + \sum_{i,j} a_i \cdot B_c^{a_j}, \tag{14}$$

where the fitted constants are $a_0$, $a_i$ and $a_j$ and $p_{nll}$ is the specific loss at the given magnetic flux density in $W/kg$. The effect of the applied technology: lamination, joints, cooling ducts in the core and the corner losses are taken by the building-factor($f_b$) into consideration, which typical value is between 1.1–1.4 [9]:

$$P_{nll} = M_c \cdot f_b \cdot p_{nll}. \tag{15}$$

The value of the applied building factor is 1.2 in the calculations.

### 2.10. Geometric Programming

A geometric program (GP) is a type of the non-linear mathematical optimization problem, characterized by the objective and constraint functions given in a special form. The name *geometric programming* refers to the geometric-arithmetic mean inequality, which used to solve GPs by the pioneers of this field [47]. The modern, fast and robust GP solvers are using interior-point methods and logarithmic change of the variables to solve these problems [22,48]:

$$\begin{aligned} &\min f_0(x) \\ &\text{s.t.} f_i(x) \leq 1, \quad i = 1, \ldots, m \\ &\quad\quad g_j(x) = 1, \quad j = 1, \ldots, n \end{aligned} \tag{16}$$

where $x = (x_1, x_2, \ldots, x_3)$ is a vector containing the optimization variables, $f_0, \ldots, f_m$ are the posynomial functions, and $g_0, \ldots, g_n$ are the monomial functions. All elements of $x$ must be positive. The monomial function $g(x)$ is a power product, it can be expressed in the following form:

$$g_j(x) = c_g \cdot x_1^{\alpha_1} \cdot x_2^{\alpha_2} \cdot \ldots \cdot x_n^{\alpha_n}, \tag{17}$$

where $c_g$ is the coefficient of the monomial and $c_g \in \mathbb{R}^+$. $\alpha_i$ is the exponent of the variable where $\alpha_i \in \mathbb{R}$. As an example, $g(x) = 3 \cdot x_1^2 \cdot x_2^{0.24} \cdot x_3^{-1.12}$ is a monomial function of the variables $x = (x_1, x_2, x_3)$.

It should be noted here that this monomial definition differs from the algebraic 'monomial' concept. In that case the exponents ($a_i$) are only non-negative integers and the coefficient is one.

The posynomial function is the sum of monomials:

$$f_i(x) = \sum_{k=1}^{l} g_k(x) = \sum_{k=1}^{K} c_k \cdot x_1^{\alpha_{1k}} \cdot x_2^{\alpha_{2k}} \cdot \ldots \cdot x_b^{\alpha_{nk}}. \tag{18}$$

where $c_k > 0$, is called a posynomial. Any monomial is also posynomial. The main advantages of GP format: firstly, this formalism guarantees that the GP solver finds the global optimum of the problem. Secondly, if the problem is infeasible this provides that no feasible point exist, the reliability and the great efficiency of the cutting edge GP solvers.

### 2.11. GP Based Embedded Winding Model

#### 2.11.1. Eddy Losses in the Windings

The objective function of this embedded geometric program to minimise the loss of the winding:

$$P_{loss} = P_{dc} + P_{ax} + P_{rad}, \tag{19}$$

$$P_{dc} = \rho \cdot \frac{2 \cdot \pi \cdot r_m}{A_{Cu}} \cdot I^2. \tag{20}$$

The load loss of the winding consist of the dc loss ($p_{dc}$) and the axial and radial components of the eddy losses ($p_{ax}, p_{rad}$). Where $\rho$ represents the specific conductivity of the conductor, and $r_m$ represents the mean radius and $I$ is the phase current of the winding. ($B_{rad}$) and ($B_{ax}$) are represents the radial and the axial components of the flux density, they are input parameters in this method, their value is calculated by the FEM part of the algorithm.

$$P_{ax} = \frac{1}{24\rho} (\omega d^* B_{ax})^2 \tag{21}$$

$$P_{rad} = \frac{1}{24\rho} (\omega h^* B_{rad})^2 \tag{22}$$

This calculation of eddy current losses in the winding segments assumes that the eddy currents do not modify the magnetic field around the winding segments [20].

#### 2.11.2. Geometry

The following posynomial inequalities and monomial constraints describe the winding arrangement, this is a disc winding with normal conductors in the examined case:

$$n = n_{ax} \cdot n_{rad}, \tag{23}$$

$$n_{ax} \cdot h + n_{ax} \cdot t_{ax} \le h_w \tag{24}$$

$$t_{hor} \cdot n_c \cdot n_{rad} + w \cdot n_c \cdot n_{rad} \le t_w, \tag{25}$$

$$A_{cu} = n_{rad} \cdot n_c \cdot w^* \cdot h^* \tag{26}$$

$$V_{cu} = 2 \cdot \pi \cdot r_m \cdot n \cdot A_{cu}, \tag{27}$$

$$\lambda_{ff} \le \frac{n \cdot A_{cu}}{h_w \cdot t_w} \tag{28}$$

$$w^* \le w_{max}, \tag{29}$$

$$h^* \le h_{max} \tag{30}$$

$$n_c \le 1, \tag{31}$$

$$n_{rad} \le 1. \tag{32}$$

where $w^*$ and $h^*$ are the searched values, the optimal width and height od a conductor. $n$ represents the number of the turns in the winding, $n_{ax}$ and $n_{rad}$ represents the axial and the radial discs in the examined case. One disc is the smallest, uniform cooling block in our case. The thermal behaviour of the whole winding can be modeled by the sum of these separate, uniform cooling blocks [3] (Figure 3). The manufacturing limits of the conductor are represented by $w_{max}$ and $h_{max}$, the $\lambda_{ff}$ is represents the filling factor, which is a lower limit in the calculation. The horizontal thickness and the axial width of the insulation is represented by $t_{hor}$ and $t_w$, and $n_c$ represents the number of conductors in a turn.

## 3. Results and Discussion

### 3.1. Validation of the Transformer Model

The accuracy and the physical correctness of the applied transformer model is demonstrated on an existing, 3 phase, 6.3 MVA, 33/22 kV, star/delta connected transformer. The core has a three-legged layout and made of M6 steel. The core filling factor was 0.85. The details of the manufactured transformer data are presented in [44].

The independent variables of the reduced transformer model is defined by the following parameters of the manufactured model:

- $D_c = 368$ mm is the core diameter,
- $B_c = 1.57$ T is the flux density,
- $h_s = 979$ mm is the height of the low voltage winding,
- $g = 26.7$ mm is the main gap distance is,
- $j_s = 3.02 \frac{A}{mm^2}$ is the current density in the LV winding,
- $j_p = 3.0 \frac{A}{mm^2}$ is the current density in the HV winding,
- $j_r = 1.86 \frac{A}{mm^2}$ is the current density in the REG.

Using these values, the optimization model gives back the same turn voltage value ($U_T = 31.0$ V) and the calculated core mass is $M_c = 4786$ kg, which is very close to the 4764 kg [44]. The high voltage winding is regulated by a linear tap changer [1]. The regulating range is 15% and the regulating winding is placed in the middle of the splitted high voltage winding (Figure 5). The main dimensions of the high voltage and the low voltage windings are depicted in Figure 5 and their main parameters—calculated and measured—are compared in Table 2.

**Table 2.** Parameters of the low and high voltage windings of the validated transformer.

|  |  | LV | | HV | |
| --- | --- | --- | --- | --- | --- |
|  |  | Reference | Model | Reference | Model |
| Line voltage | kV | 22 | | 35 | |
| Connection | kV | D | | Y | |
| Phase Voltage | kV | 22 | | 20.23 | |
| Number of turns | # | 708 | | 650 | |
| Phase current | A | 95.5 | | 104 | |
| Turn area | mm$^2$ | 31.623 | | 56.0 | |
| Conductor height | mm | 11.6 | 6.6 | 11.4 | 8.1 |
| Conductor width | mm | 2.7 | 2.7 | 3 | 2.7 |
| Mean diameter | mm | 437 | 436 | 578 | 572 |
| Winding width | mm | 42.9 | 42.8 | 40.7 | 41.1 |
| Copper mass | kg | 813 | 824 | 1071 | 1082 |
| Loss | kW | 19.150 | 19.23 | 25.948 | 23.979 |

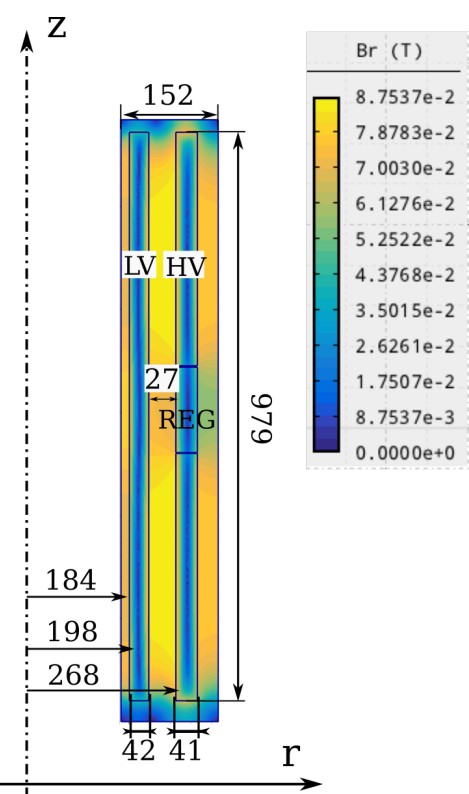

**Figure 5.** Main dimensions and the flux density distribution of the validation example and the main parameters in Agros2D [34].

It can be seen from the results that the calculated losses are very close to the reference values. The resulting losses of the optimization are smaller, this can be the result of the applied methodology, which found different conductor heights for the optimum. The difference between the radial width of the windings is not significant, it is lesser than half of the mm. This can happen, because the outline sizes of the windings are calculated by the usage of the winding filling factors, which not differentiates in the radial and in the axial direction. However, the filling factor is smaller in the axial direction, because of the applied cooling duct heights between the discs. The calculated short-circuit impedance (SCI) is 7.43%, which is very close to the detailed model based calculations (7.18%) [44].

*3.2. Input Parameters of the Test Transformer*

The optimization method was tested on the following case study: a cost optimization of a 31.5 MVA power transformer with 132 kV/33 kV voltage ratio. The objective of the optimization is the total cost of the ownership. The network frequency is 50 Hz, the required short circuit-impedance is 14.5%. The parameters are selected according to the standard [4]. The TOC is calculated by the following capitalization factors: $K_1$ = 6000 €/kW and $K_2$ = 2000 €/kW. For the sake of simplicity, the transformer cooling was chosen to be ONAN and the ambient temperature was specified to 40 °C. The allowed winding oil temperature rise was defined to $\Theta_{wo}$ = 65 K, according to the IEC-60076 standard [49]. Therefore, the winding current density limit was set to 3.0 A/mm$^2$ in the main windings and 3.5 A/mm$^2$ in the regulating winding. The primary winding was modeled as a helical winding from CTC, while the secondary winding as a disc winding from twin conductors. The transformer is regulated by a diverter switch, which switch off the regulating winding at the nominal tapping stage. The applied core material in this case was a TRAN-COR H1 grade electrical steel. The maximum value of the flux density was limited to 1.7 T considering the saturation of the core material and over-voltages in the power grid. The minimal insulation distances were chosen by empirical rules [9]. These methods were based on the lightning impulse test and AC test requirements. The detailed list of the input

parameters of the optimization model are presented in Table 3. The upper and the lower bounds of the searched independent parameters are presented in Table 4.

**Table 3.** List of the optimization model input parameters.

| Parameter | | Dimension | Value |
|---|---|---|---|
| Nominal power | | MVA | 31.5 |
| Frequency | | Hz | 50 |
| Connection group | | | Dyn1 |
| Number of phases | | # | 3 |
| Short circuit impedance | | % | 14.5 |
| Main gap | | mm | 37 |
| Sum of the end insulation | | mm | 150 |
| Phase distance | | mm | 37 |
| Core-Inner winding distance | | mm | 20 |
| | Number of legs | # | 3 |
| | Flux density limit in columns | T | 1.7 |
| Core | Filling Factor | % | 90 |
| | Material Type | | M1H |
| | Material Price | €/kg | 3.5 |
| | Line Voltage | kV | 33 |
| | Phase Voltage | kV | 19.05 |
| Low Voltage | BIL | kV | 125 |
| Winding | AC | kV | 50 |
| | Copper filling factor | % | 60 |
| | Material and manufacturing price | €/kg | 10 |
| | Line Voltage | kV | 120 |
| | Phase Voltage | kV | 69.36 |
| High Voltage | BIL | kV | 550 |
| Winding | AC | kV | 230 |
| | Copper filling factor | % | 60 |
| | Material and manufacturing price | €/kg | 8.5 |
| | Regulating range | % | ±10 |
| Regulating | Insulation | | Fully insulated |
| Winding | Regulated winding | | High voltage |
| | Filling factor | % | 65 |

**Table 4.** The applied bounds for the independent parameters.

| Parameter | Dimension | Lower Bound | Upper Bound |
|---|---|---|---|
| $D_c$ | mm | 400 | 700 |
| $B$ | T | 1.4 | 1.7 |
| $g$ | mm | 37 | 70 |
| $j_s$ | A/mm$^2$ | 1.5 | 3.0 |
| $j_p$ | A/mm$^2$ | 1.5 | 3.0 |
| $j_r$ | A/mm$^2$ | 1.5 | 3.5 |
| $h_s$ | mm | 1200 | 2000 |

## 3.3. Discussion of the Results

The main goal of this task is to verify that the resulted TOC of the proposed algorithm can find the global optimum of the equation system. The result of the cost optimization are the TOC and the key-design parameters of the transformer (Figure 6). The key-design parameters are good for a design study and a cost estimation, but these parameters can not determine one and only final transformer design. Therefore, the result of the proposed method (Table 5) are compared with the results of a previously validated, metaheuristic based optimization method. The metaheuristic method uses the combination of the method of branch and bound and geometric programming to find the

optimal solution of an analytical transformer model. It was shown in a previous article [27], that the usage of this geometric programming based solver guarantees that this method finds the global optima of the optimization task [14,27]. The physical correctness of the results of the metaheuristic method were validated by FEM. However, the robustness and the precision of the used analytical formulas is about 5% lesser, compared to the FEM based calculations [14,27]. This difference explains the relatively small difference between the optimal values of the two TOCs. This difference is relatively small and acceptable, lesser than 1%. Figure 7 illustrates the convergence of the algorithm. During the optimization, the NSGA-II algorithm was generated 100 individuals for every 100 generations. As it can be seen on Figure 7, after 80th iteration the algorithm was found the optimal solution. However, the shapes of the two resulted transformer designs are very different. The main reason of this difference is the non-linearity of the transformer design optimization problem and the differences between the mathematical representation of the problem. These significant differences in the key-design parameters indicates that it is important to use the proposed, extended model to find more robust solutions.

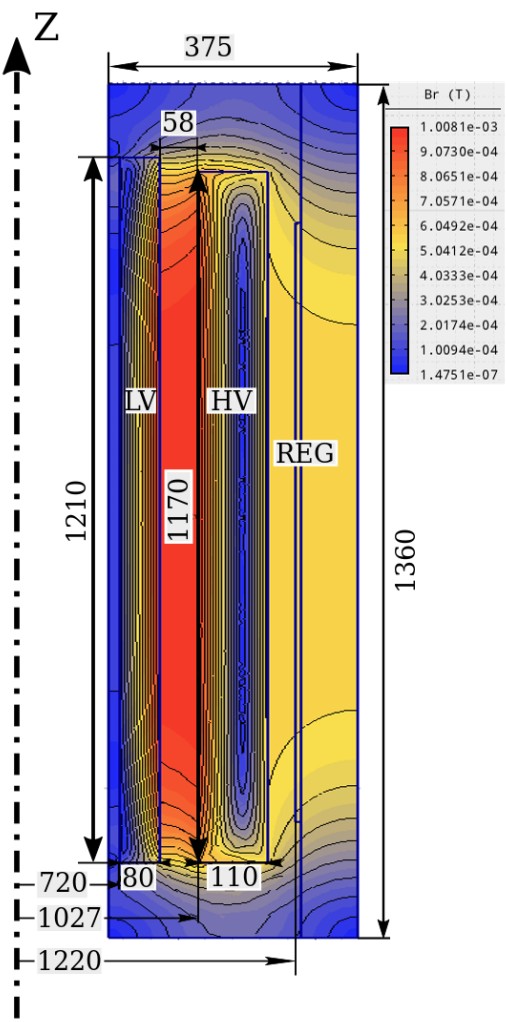

**Figure 6.** Results of the optimization process, namely optimal key-design parameters and magnetic flux distribution in window of optimized, 31.5 MVA, 132 kV/33 kV power transformer. The FEM calculation made by Agros2D [34]. Symbol LV means the low voltage winding, symbol HV represent the modeled high voltage winding.

**Table 5.** List of the optimization model results.

| Design Parameters | Dimension | Metaheuristic | NSGA2+GP |
|---|---|---|---|
| **Core data** | | | |
| core diameter | mm | 570 | 600 |
| flux density | T | 1.64 | 1.58 |
| core mass | t | 16.65 | 21.05 |
| turn voltage | V | 83.6 | 89.3 |
| main gap | mm | 37 | 58 |
| **Low voltage winding** | | | |
| inner diameter | mm | 610 | 720 |
| winding height | mm | 1003 | 1210 |
| winding width | mm | 89 | 80 |
| turn number | # | 228 | 214 |
| current density | A/mm$^2$ | 2.35 | 2.02 |
| h* | mm | - | 3.6 |
| w* | mm | - | 2.5 |
| **High voltage winding** | | | |
| inner diameter | mm | 861 | 1027 |
| winding height | mm | 973 | 1170 |
| winding width | mm | 107 | 110 |
| turn number | | 1579 | 1478 |
| h* | mm | - | 8.1 |
| w* | mm | - | 2.7 |
| current density | A/mm$^2$ | 2.01 | 1.53 |
| **Regulating winding** | | | |
| inner diameter | mm | 1149 | 1220 |
| winding height | mm | 853 | 1025 |
| winding width | mm | 10 | 10 |
| current density | A/mm$^2$ | 2.7 | 2.71 |
| load loss | kW | 114.9 | 88.3 |
| core loss | kW | 13.2 | 17.82 |
| TOC | € | 447,627 | 448,597 |

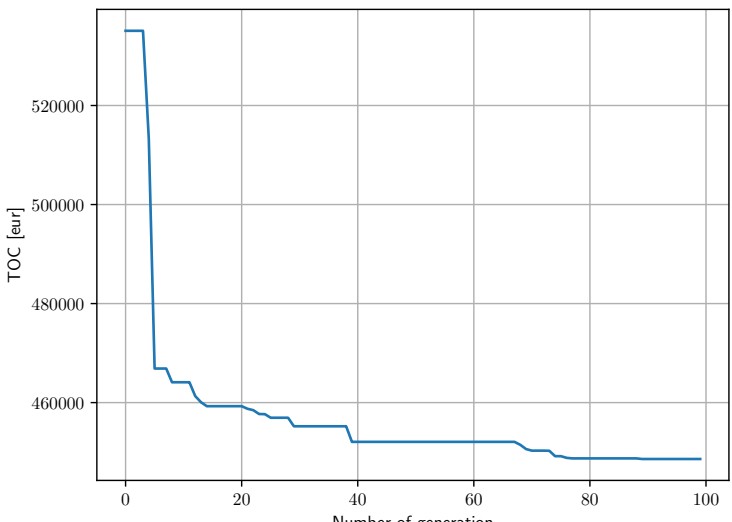

**Figure 7.** Evolution of the TOC during the optimization process.

The most significant difference is found in the main gap selection. The metaheuristic method was found a solution, where the length of the main gap is equals with the possible minimum (37 mm), according to the practical design rules. However, the proposed method was yielded the cheapest solution with a much larger main insulation distance (58 mm). This difference shows also the non-linearity of the task: a small difference between the optimized TOC values can lead a significant difference in the cost optimal key-design parameters and the minimisation of the insulation distances not leads automatically a cheaper design. The optimal conductor sizes in the case of the HV winding are very realistic, these results verify the applicability of the method. The optimal conductor shape values of the LV winding corresponds with the theoretical expectations, however they are not so realistic. The reason of this problem, that the thermal and the mechanical properties of the winding are still not considered during the calculation. The relatively small copper height and the cubic shape of the conductor leads to a wrong thermal behaviour and manufacturability. Therefore, the thermal and the mechanical properties should be considered in the future to get a more practical solution.

## 4. Conclusions

This paper has proposed an algorithm, which can determine the optimal conductor sizes for the transformer windings. This algorithm uses evolutionary algorithm (NSGA-II) based search to find the optimal key-design parameters of the simplified transformer model. This simplified transformer model uses a FEM calculation directly in the sole calculation loop to determine the short-circuit impedance and the magnetic field distribution in the working window of the transformer . From the knowledge of the winding shape and the magnetic flux density, a geometric programming based method is used to calculate the optimal winding layout and conductor shapes. The application of geometric programming ensures that the optimal solution is exist and the found optimum is the global optimum. The applied algorithm can successfully use the widely used and precise FEM based formulas [20] for load loss calculation from the optimal core shapes. This study has shown, that the optimal conductor sizes can be estimated, so the thermal properties of a transformer can be considered at the beginning of the design. The presented test example has shown, that the proposed algorithm can find the optimal solution in reasonable computation time. The result of the goal function corresponds with the result of a well tested, metaheuristic transformer optimization method. The main advantages of using this method with the proposed optimization framework, that it can find more robust solutions and it can be easily extendable with other quantities in the future. A further research can extend this method with the winding temperature calculations to analyse the impact of the application of different cooling systems, insulation liquids on the cost optimal parameters.

**Author Contributions:** Conceptualization, T.O.; Funding acquisition, P.K.; Methodology, T.O.; Resources, P.K.; Software, D.P.; Supervision, D.P. and P.K.; Visualization, D.P.; Writing—original draft, T.O. All authors have read and agreed to the published version of the manuscript.

**Funding:** This research was funded by the Ministry of Education, Youth and Sports of the Czech Republic under the RICE New Technologies and Concepts for Smart Industrial Systems, project No. LO1607 and by an internal project SGS-2018-043.

**Conflicts of Interest:** The authors declare no conflict of interest.

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
