# Peer review of "FEM Based Preliminary Design Optimization in Case of Large Power Transformers"

_applsci, doi:10.3390/app10041361_

Round 1

Reviewer 1 Report

Although the set-up of the paper is absolutely good with a right introduction that describes the problem, a correct tailored Mathematical section, a good  test one and a good bibliography, the paper is not exciting. Nevertheless, the overall quality is good and shows a certain grade of innovation, therefore, for my side, it can be accepted in this form. 

being the paper extremely operative with many figures and tables which show the results, it is difficult to suggest something to improve because the verbose section is correct and essential. I give to it an average evaluation because everything is correct but nothing is exceptional including the innovation.

The bibliography is correct, the paper is well written, the Mathematical section is correct and with a right size, the simulations are correct and give good results that confirm the theoretical approach, but nothing is exciting.

Therefore, I suggest to accept the paper in this form, but unfortunately I can't suggest anything that is compatible with the time of a revision. 

Reviewer 2 Report

This paper presents a hybrid approach to optimize layout of windings to obtain the minimum of the total cost of ownership (TOC). Authors propose to use analytical formula, FEM programs, and Non-dominated Sorting Genetic Algorithm in iteration loop to obtain convergent of results. I appreciate the authors' approach, but I have a number of very critical comments listed below in a general list and in a detailed list.

General comments:

What are simplifying assumptions of a transformer model? They should be listed.

How do simplifying assumptions influence the objective function?

FEM calculations are usually time consuming process especially in iteration loops. What is the time of FEM calculations with respect to the time utilised by other methods used for similar problem?

What type of FEM calculations was used: magnetostatics, AC steady state, AC transients etc? What software for FEM calculation was used? There is no information.

What value of relative permeability was used for FEM calculations? B-H curve is strongly nonlinear and value of relative permeability can be in the range of thousands to several dozen. What is the influence of relative permeability on the TOC?

FEM simulations were performed for short circuit condition. How is equilibrium of magneto motive forces of primary, secondary and regulation windings implemented in FEM simulations?

The objective function have particular costs (manufacturing, active parts, and dissipated energy). What are the percentage proportions of these particular costs in relation to TOC?

How eddy current loss, hysteresis loss, and anomalous losses dissipated in transformer core are calculated? Is important for TOC?

How skin effect and proximity effect in transformer windings are taken into account? Is important for TOC?

Detailed comments:

I am asking for a detailed explanation of this sentence: (Page 2, row 47, 48) "Because of the strong non-linearity and the nature of the problem, there is a small difference between the global cost optima and some other local optimums".

Shape of the magnetic core and windings (Fig.2) are not visible. Cross section view is also recommended.

Magnetic flux density was assumed the same for column and yoke. In real transformer they are different. I am asking the authors for a comment on this matter.

What is the difference betwen n and nc in the Table 1?

There are errors in equation 3

In the equation 6 not all components are described.

Round 2

Reviewer 2 Report

General comments:

What are simplifying assumptions of a transformer model? They should be listed. There is no a list of simplifying assumptions. What about: B-H curve (linear/nonlinear), structure of magnetic core (homogenized/laminated silicon steel), interleaved joints of core (interleaved/air gaps), arrangement of turns in windings, etc.

How do simplifying assumptions influence the objective function? Answer is not satisfied. Results of a few FEM calculations at different relative permeability should be used to calculate the TOC and described in the paper.

FEM calculations are usually time consuming process especially in iteration loops. What is the time of FEM calculations with respect to the time utilized by other methods used for similar problem?

I accept explanation given by the Authors.

What type of FEM calculations was used: magnetostatics, AC steady state, AC transients etc.?

I accept explanation given by the Authors.

What software for FEM calculation was used? There is no information.

I accept explanation given by the Authors.

What value of relative permeability was used for FEM calculations? B-H curve is strongly nonlinear and value of relative permeability can be in the range of thousands to several dozen. What is the influence of relative permeability on the TOC?

Answer is not satisfied. There are no details about: B-H curve, value of the relative permeability, and how this affects the TOC value. Value of relative permeability and its meaning on the TOC value should be discussed in the paper.

FEM simulations were performed for short circuit condition. How is equilibrium of magneto motive forces of primary, secondary and regulation windings implemented in FEM simulations?

Answer is not satisfied. Magneto motive force (MMF) is the product of number of turns and current in winding MMF = N*I. Each transformer column has 3 windings: primary with Np turns and Ip current, secondary with Ns turns and Is current, and regulation with Nr turns and Ir current.If magnetizing current Im is omitted the equilibrium of MMFs is given by: Np*Ip+Ns*Is+Nr*Ir=0. The question is: Wat are values of Np, Ip, Ns, Is, Nr, Ir in FEM calculations presented in the paper? These values should be given in the paper. Explanation: magnetomotive force is not the same as electromagnetic force

The objective function have particular costs (manufacturing, active parts, and dissipated energy). What are the percentage proportions of these particular costs in relation to TOC?

I accept explanation given by the Authors

How eddy current loss, hysteresis loss, and anomalous losses dissipated in transformer core are calculated? Is it important for TOC?

Answer is not satisfied. What are eddy current loss, hysteresis loss, and anomalous losses in transformer core calculated by Authors for analyzed transformer? Are they omitted?

How skin effect and proximity effect in transformer windings are taken into account? Is it important for TOC?

In my opinion equations (16) and (17) don’t take into account the proximity effect. Is it omitted?

I accept corrections - regarding my detailed comments - given by the Authors.

Author Response

What are simplifying assumptions of a transformer model? They should be listed. There is no a list of simplifying assumptions. What about: B-H curve (linear/nonlinear), structure of magnetic core (homogenized/laminated silicon steel), interleaved joints of core (interleaved/air gaps), arrangement of turns in windings, etc.

A large power transformer is not equals with its active part, it has many, non-electrical assemblies. This simplification is the most important and written in the beginning of the section. Then the section introduces the differences between the current and the existing methods. These simplifications are inserted and written in the different part of the section, not a list format, but the text contains all of the above mentioned information:

The core loss calculation: rewritten, then its placed at the end of the analytical subsection.

The structure of the magnetic core – also placed in the analytical part, but a short description added to the beginning of the text.

Modelling the arrangement of turns in windings – a small explanation added to the beginning of the text, because the previous preliminary design optimization methods in case of large power transformers were used a copper filling factor-based estimation for the different winding types.

How do simplifying assumptions influence the objective function? Answer is not satisfied. Results of a few FEM calculations at different relative permeability should be used to calculate the TOC and described in the paper.

Using different relative permeabilites is not necessary in case of power transformers, as written in the book Kulkarni, Transformer Engineering, page 92 in the first edition of the book (2004)] to fine tune this calculation.

The magnetic core calculation described in more details, the detailed text gives the answer for this question:

The magnetic core can be defined by its relative permeability mu_r, it can be some of tens of thousands. During the simulations it was defined as mu_r= 10 000. It can be a number between 10000 and 50000, as stated in [Kulkarni, Transformer Engineering, page 92 in the first edition of the book (2004)]. However, it doesn't effect on the solution, because almost all energy is stored in the non-magnetic regions, where mu_r = 1, outside of the core. We can also use the assumption of [20], that the radial component of the magnetic flux density is perpendicular to the core. Other regions, including the windings are defined by mu_r = 1.

FEM calculations are usually time consuming process especially in iteration loops. What is the time of FEM calculations with respect to the time utilized by other methods used for similar problem?

I accept explanation given by the Authors.

What type of FEM calculations was used: magnetostatics, AC steady state, AC transients etc.?

I accept explanation given by the Authors.

What software for FEM calculation was used? There is no information.

I accept explanation given by the Authors.

What value of relative permeability was used for FEM calculations? B-H curve is strongly nonlinear and value of relative permeability can be in the range of thousands to several dozen. What is the influence of relative permeability on the TOC?

Answer is not satisfied. There are no details about: B-H curve, value of the relative permeability, and how this affects the TOC value. Value of relative permeability and its meaning on the TOC value should be discussed in the paper.

The text changed and discussed in more details and gives the answer for permeability, effects on TOC:

The magnetic core can be defined by its relative permeability mu_r, it can be some of tens of thousands. During the simulations it was defined as mu_r= 10 000. It can be a number between 10000 and 50000, as stated in [3] (page 92 in the first edition of the book (2004)) However it doesn't effect on the solution, because almost all energy is stored in the non-magnetic regions, where mu_r = 1, outside of the core. We can also use the assumption of [20], that the radial component of the magnetic flux density is perpendicular to the core. Other regions, including the windings are defined by mu_r = 1.

FEM simulations were performed for short circuit condition. How is equilibrium of magneto motive forces of primary, secondary and regulation windings implemented in FEM simulations?

Answer is not satisfied. Magneto motive force (MMF) is the product of number of turns and current in winding MMF = N*I. Each transformer column has 3 windings: primary with Np turns and Ip current, secondary with Ns turns and Is current, and regulation with Nr turns and Ir current.If magnetizing current Im is omitted the equilibrium of MMFs is given by: Np*Ip+Ns*Is+Nr*Ir=0. The question is: Wat are values of Np, Ip, Ns, Is, Nr, Ir in FEM calculations presented in the paper? These values should be given in the paper. Explanation: magnetomotive force is not the same as electromagnetic force

The balance of the amperturns are checked at the beginning of the FEM calculation. At that stage the code uses the copper filling factor * winding area * current density equation to calculate the magnetomotive force, this data is given in Table 5. What is missing for you that is not stated that the regulation winding is modeled in a de-energized state, because it is a diverter switch, which usually switched off at the nominal tapping stage, so Nr * Ir = 0 at that stage, NpIp = 117937

The objective function have particular costs (manufacturing, active parts, and dissipated energy). What are the percentage proportions of these particular costs in relation to TOC?

I accept explanation given by the Authors

How eddy current loss, hysteresis loss, and anomalous losses dissipated in transformer core are calculated? Is it important for TOC?

Answer is not satisfied. What are eddy current loss, hysteresis loss, and anomalous losses in transformer core calculated by Authors for analyzed transformer? Are they omitted?

The section of the core loss calculation is completely rewritten in the end of the analytical formulas section.

How skin effect and proximity effect in transformer windings are taken into account? Is it important for TOC?

In my opinion equations (16) and (17) don’t take into account the proximity effect. Is it omitted?

Yes, it is omitted, it is written in the paper on page 9 and line 194-195 and in the cited source of the method [20]. “This calculation of eddy current losses in the winding segments assumes that the eddy currents do not modify the magnetic field around the winding segments [20 ].”

The accuracy of this method is higher than the previously used analytical formulas, this is a well-known and widely used method in the transformer industry.

Round 3

Reviewer 2 Report

All critical aspects are explained